



# Surface mass balance and water stable isotopes derived from firn cores on three ice rises, Fimbul Ice Shelf, Antarctica

Carmen P. Vega,[1,2] Elisabeth Schlosser,[3,4] Dmitry V. Divine,[1] Jack Kohler,[1] Tõnu Martma,[5] Anja Eichler,[6] Margit Schwikowski[6], and Elisabeth Isaksson[1]

[1]Norwegian Polar Institute, N-9296 Tromsø, Norway
[2]Department of Earth Sciences, Uppsala University, Villavägen 16, SE 752 36, Uppsala, Sweden
[3]Institute of Atmospheric and Cryospheric Sciences, University of Innsbruck, Innsbruck, Austria
[4]Austrian Polar Research Institute, Vienna, Austria
[5]Institute of Geology, Tallinn University of Technology, Tallinn, Estonia
[6]Paul Scherrer Institute, 5232 Villigen PSI, Switzerland

*Correspondence to*: Carmen P. Vega (carmen.vega@geo.uu.se)

**Abstract.** Three shallow firn cores were retrieved in the austral summers of 2011/12 and 2013/14 on the ice rises Kupol Ciolkovskogo (KC), Kupol Moskovskij (KM), and Blåskimen Island (BI), all part of the Fimbul Ice Shelf (FIS) in western Dronning Maud Land (DML), Antarctica. The cores were dated back to 1958 (KC), 1995 (KM) and 1996 (BI) by annual layer-counting using high-resolution oxygen isotope ($\delta^{18}$O) data, and by identifying volcanic horizons using non-sea salt sulphate (nssSO$_4^{2-}$) data. The water stable isotope records show that the atmospheric signature of the annual snow accumulation cycle is well preserved in the firn column, especially at KM and BI. We are able to determine the annual surface mass balance (SMB), as well as the mean SMB values between identified volcanic horizons. Average SMB at the coastal KM and BI sites (0.68 m w.e. yr$^{-1}$ and 0.70 m w.e. yr$^{-1}$) was higher than the more inland KC site (0.24 m w.e. yr$^{-1}$), and there was greater temporal variability as well. Trends in the SMB and $\delta^{18}$O records from the KC core over the period of 1958–2012 agree well with other previously investigated cores in the area and thus the KC site could be considered as the most representative of the climate of the region. Cores from KM and BI appear to be more affected by local meteorological conditions and surface topography. Our results suggest that the ice rises are suitable sites for the retrieval of longer firn and ice cores, but that BI has the best preserved seasonal cycles of the three records and is thus the most optimal site for high-resolution studies of temporal variability of the climate signal. Deuterium excess data suggests a possible role of seasonal moisture transport changes on the annual isotopic signal. In agreement with previous studies, large-scale atmospheric circulation patterns most likely provide the dominant influence on water stable isotope ratios preserved at the core sites.

## 1 Introduction

The Antarctic ice sheet plays a major role in the global climate system; nevertheless, despite much recent attention, there are still many unresolved issues around both its mass balance and recent climate history, particularly in East Antarctica (IPCC,



2013). Estimating mass balance for the ice sheet from field data is made difficult by the logistical challenges of collecting in situ data, as well as the enormous size of the region. Satellite methods are compromised by the fact that most of the region is close to equilibrium. Even when combining several different methods, corrections for isostatic rebound and changes in firn density, relatively poorly known quantities in Antarctica can alter the overall mass balance estimate from positive to negative

(i.e. Shepherd et al., 2012; Zwally et al., 2015). Therefore, given the future projections of greenhouse gas emissions and the associated temperature rise, the onset of a possible significant contribution of Antarctica to sea level rise is difficult to predict with any high accuracy (e.g. IPCC, 2013; DeConto and Pollard, 2016).

While the interior of the continent contains most of the ice volume, the coastal regions are the most vulnerable part of Antarctica with regards to climate warming. In addition to increasing atmospheric temperatures, changes in storm tracks and

10 the impact of warmer ocean currents penetrating further south, all will impact future behaviour of the coastal ice.

The ice shelves surrounding Antarctica stabilize the grounded interior ice (e.g. Vaughan and Doake, 1996). There has been significant thinning and even disintegration of ice shelves during the last decades (e.g. Scambos et al., 2004; Shepherd et al., 2010; Pritchard et al., 2012; Paolo et al., 2015), leading to increased outflow of glaciers and ice streams that feed the shelves. Warmer ocean water has been identified as important to the ice shelf removal (e.g. Pritchard et al., 2012), highlighting the

15 importance of the ice-ocean interactions, particularly at the grounding zone.

Ice rises and ice rumples are elevated small-scale topographic features on ice shelves, areas of grounded ice surrounded by floating ice. They buttress the ice shelves and represent an important part of the ice sheet complex (Paterson, 1994; Matsuoka et al., 2015). Ice flow on ice rises is typically independent of the surrounding ice shelf, with radial flow due to their dome-like morphology. Furthermore, ice velocities are generally low on ice rises; this fact, together with their relatively

high surface mass balance (SMB) due to their location at the coast, make ice rises potentially useful sites for ice core studies. There are numerous ice rises along the rim of the Antarctic continent and few of them have been studied for the purpose of ice core drilling. For more details on ice rises we refer to a recent review paper by Matsuoka et al. (2015).

Antarctic ice and firn cores contain valuable information about the climate and chemical composition of the atmosphere. Numerous ice and firn cores have been drilled in Antarctica during the past decades. Ice core studies typically focus either

on long timescales, such as the EPICA, Vostok, Dome Fuji, and WAIS Divide projects (e.g. Watanabe et al. 1999; EPICA community members, 2006; Wolff et al., 2010; WAIS Divide Project Members, 2013), or on spatial distribution of climate and glaciological parameters, e.g. within projects such as ITASE (Mayewski et al., 2005). Most studies are on ice cores drilled in the dry interior of Antarctica, where the SMB is low; there are far fewer studies of ice core records from the coastal regions, which are more sensitive to climatic changes than the interior of the continent. The higher SMB of coastal sites

allows high-resolution records to be obtained, thus providing the possibility of comparing firn or ice core data to instrumental records available since the middle of the 20[th] century (e.g. Schlosser et al., 2014).

The primary overall goal of the project *Ice Rises* is to elucidate the mass-balance history of three ice rises in a section of Dronning Maud Land (DML) (Figure 1) over the past several millennia. Understanding the past changes in their SMB,



specifically during past warm anomalies, will eventually help to improve the understanding of the impact of the predicted future atmospheric and oceanic warming on the mass balance of the Antarctic ice sheet.

During two Antarctic field seasons, in the austral summer of 2011/12 and 2013/14, a number of glaciological field data have been collected at three ice rises located in the Fimbul Ice Shelf (FIS): Kupol Ciolkovskogo (KC), Kupol Moskovskij (KM),

and Blåskimen Island (BI) (Figure 1). In this paper we focus on the SMB and water isotope records obtained from these cores, with emphasis on differences between the sites to evaluate their representativeness for the area. These data are an important input to mass balance models and to assess the suitability of these coastal sites as possible drill location for deeper ice core retrieval.

## 2 Field area

The Fimbul Ice Shelf (Figure 1) is one of many ice shelves along the coast of DML. It measures roughly 36 500 km$^2$ and is the largest ice shelf in the Haakon VII Sea. FIS is fed by the fast-flowing ice stream Jutulstraumen, whose ice velocity is ~1 km yr$^{-1}$ at the grounding line, some 200 km inland from the shelf edge (Melvold and Rolstad, 2000; Rolstad et al., 2000). Jutulstraumen is the largest outlet glacier in DML, draining an area of 124 000 km$^2$, and is therefore important to the mass balance of the ice shelf. FIS is comprised of a fast-moving part that extends from Jutulstraumen and protrudes into the sea,

Trolltunga, surrounded by the slower-moving ice shelf proper. Trolltunga extends north across the narrow continental shelf separating the glaciated coast from the warm water of the coastal oceanic current, making it potentially vulnerable to basal melting (e.g. Hatterman et al., 2012).

A number of ice rises varying in size from 15 to 1200 km$^2$ are found in the ice shelf. The three ice rises investigated in this study are situated approximately 200 km from each other (Figure 1). All ice rises are dome-shaped with elevations ranging

from 260–400 m a.s.l. Ice radar studies at the core sites suggest ice depths from 350–460 m; ice velocities from GPS measurements show values in the order of 2 m yr$^{-1}$ or less at the core sites (Brown, Goel and Matsuoka, personal communication). The northern edges of KM and BI border the ocean, while KC is surrounded by the ice shelf (Figure 1). During three field seasons (2011/12, 2012/13 and 2013/14), radar, ice velocity, and stake data were collected, with the overall goal of studying ice rise mass balance evolution over time. Preliminary data analysis suggest that ice velocities across

the ice rises are asymmetrical and that the surface mass balance distribution is variable over the three ice rises (Brown et al., 2014).

The SMB of the ice rises is influenced by precipitation, wind erosion and redeposition, and by sublimation from the surface and from drifting snow. FIS, like most East Antarctic ice shelves, is under the climatic influence of the circumpolar trough; precipitation comes mainly from frontal systems of cyclones moving eastwards, north of the coast, resulting in easterly or

east-north-easterly surface winds (Schlosser et al., 2008). These events occur frequently, and throughout the year. Precipitation amounts during an event depend on the temperature and humidity of the involved air masses, with moisture transport from lower latitudes leading to higher precipitation amounts than cyclogenesis in the polar ocean. However, the




local meteorological conditions at the ice rises differ from the rest of the ice shelf: air temperatures are higher due to a weaker temperature inversion in winter, and wind speeds are higher due to the fact that the ice rises represent obstacles in the general atmospheric flow (Lenaerts et al, 2014). Studies have shown that the relative height of the obstacle compared to its horizontal dimensions, the wind speed, and the static stability of the atmosphere determine whether there is more precipitation on the windward or lee side of the obstacle (Rotunno and Houze, 2007; Houze, 2012). This refers to precipitation alone; redistribution of snow by wind can strongly influence the SMB of the ice rises and consequently, large differences in SMB are expected to be found over relatively short distances close to the ridge of the ice rise.

## 2.1 Previous work

The first scientific work in the study area was conducted during the British-Norwegian-Swedish Expedition in 1949–1952, which made detailed descriptions of both the geology and morphology of the Jutulstraumen basin, including the ice sheet and ice shelf (Swithinbank, 1957). Work in the area was continued during the International Geophysical Year (IGY) 1956/57, at the Norway Station (later re-named SANAE), on the western edge of the ice shelf between 1956 and 1960 (Lunde, 1961; Neethling, 1970). In the last three decades Norwegian groups have worked on FIS and Jutulstraumen under the auspices of the Norwegian Antarctic Research Expedition (NARE), focusing on spatial and temporal variability of SMB using shallow firn cores and Ground Penetrating Radar (GPR) (e.g., Melvold et al., 1998; Melvold, 1999; Isaksson and Melvold, 2002; Sinisalo et al., 2013; Schlosser et al., 2012 and 2014).

As part of the EPICA project, a 100-m deep ice core (labelled S100) was drilled on the eastern part of FIS during NARE 2000/2001 (Figure 1). This core covers the period 1737–2000 A.D. ± 3 years and shows higher SMB values in the nineteenth century than in the eighteenth and twentieth century, but otherwise no significant trends (Kaczmarska et al., 2004). This core is the longest available high-resolution climate record from this part of coastal DML.

Rotschky et al. (2007) compiled a SMB map for western DML, including FIS, but data were not available to resolve fine-scale variability in the area of the ice rises. More recently, Sinisalo, et al. (2013) and Lenaerts et al. (2014) using field and model data show that the ice rises have a substantial role in shaping both local SMB and meteorological conditions. Finally, Altnau et al. (2015) compile available oxygen stable isotopes and SMB data for the last three decades; they find a negative SMB trend for the coastal regions, but a positive trend on the polar plateau over the same time period. They conclude that atmospheric dynamic effects are more important at the coast than thermodynamics, the latter being the dominant factor on the polar plateau, where changes in SMB and stable isotope ratios occur mostly in parallel.

## 3 Methods

### 3.1 Sampling

Three shallow (ca. 20 m) firn cores were retrieved at FIS (Figure 1, Table 1) in January 2012 (KC) and January 2014 (KM and BI) during field expeditions organized by the Norwegian Polar Institute (NPI). Table 1 presents the location, elevation



and length of the different ice rises cores. Each core was drilled from the bottom of a 2 m snow pit; the pit wall was sampled at 5 cm intervals for water stable isotope analysis. Bulk core density was determined for each sub-core piece (average length ~45 cm) and for each snow pit sample (20 cm). Snow and firn samples were collected following clean protocols (Twickler and Whitlow, 1997), shipped frozen to NPI, and later to the Paul Scherrer Institute (PSI), Switzerland, for cutting and

chemical analysis. Sample length ranged from 4–8 cm, depending on the sample depth and density. The presence and thickness of ice lenses were recorded during cold room analysis of the KC core. Major ions (MSA, $SO_4^{2-}$, and $Na^+$) were analysed at PSI. Sub-samples for water stable isotopes analysis were shipped to the Institute of Geology at Tallinn University of Technology (TUT), Estonia.

### 3.2 Water stable isotopes and major ion analyses

Water stable isotope ratios ($^{18}O/^{16}O$ and $^2H/^1H$) were measured at TUT using a Picarro L2120-i water isotope analyser (cavity ring-down spectroscopy technology) with a high-precision A0211 vaporizer. Measurements were calibrated against both VSMOW and the Vienna Standard Light Antarctic Precipitation (VSLAP) standards. Reproducibility of $\delta^{18}O$ and $\delta^2H$ measurements was ± 0.1 ‰ and ± 1 ‰ (for 4–6 replicate measurements), respectively. Measuring both oxygen and hydrogen water stable isotopes in the ice rises cores yields deuterium excess ($d = \delta^2H - 8\delta^{18}O$).

Major ions (MSA, $SO_4^{2-}$, and $Na^+$, Table 2) were analysed at PSI using a Metrohm ProfIC 850 ion chromatography system combined with an 872 Extension Module and auto-sampler. The precision of the method was around 5 % for all ions (Wendl et al., 2015). In this study we use records of major ions $Na^+$ and MSA to corroborate the dating of the three cores (see 4.1.), performed by identifying seasonal cycles in the oxygen isotope record; a detailed paleoenvironmental analysis at the ice rise sites using the ion data is the subject of a separate paper in progress.

**4 Results**

### 4.1 Dating of the firn cores

Due to higher accumulation rates, the $\delta^{18}O$ seasonal variability in the coastal cores KM and BI (Figure 2b and c) is better defined than at the more inland KC site, and dating uncertainty for the coastal cores is therefore lower. Dating of the firn cores is performed by annual layer-counting, using the seasonality of the water stable isotope signal. Since the KM and BI

cores were drilled from the bottom of a snow pit (0.9 m w.e.), the snow pit data are used to reconstruct the period between winter-2012 and summer-2014. Winter minima and summer maxima in the $\delta^{18}O$ record are identified to obtain a timescale with sub-annual resolution. Assuming a uniform distribution of precipitation throughout the year, an equidistant timescale is adapted between the summer maxima (January) and the winter minima (July). Well pronounced seasonal cycles of major ion concentrations (e.g. MSA and $Na^+$, Figure 3) are used to corroborate the dating. Based on annual layer counting, the KM and



BI cores cover the periods between winter 1995 to summer 2014, and winter 1996 to summer 2014, respectively. The error in the dating is estimated as ± 1 year for both of these cores.

Counting $\delta^{18}$O winter minima in the KC core is not as straightforward as for the KM and BI cores, due to the lower accumulation and the lower amplitude of the seasonal signal (Figure 2a). Using $\delta^{18}$O snow pit data available for the surface

layers at KC (Figure 2a), a SMB of 0.19 m w.e. yr$^{-1}$ is estimated for the period 2007–2011. Accordingly, when interpreting the seasonal variability of the $\delta^{18}$O stratigraphy, this mean SMB 2007–2011 value was considered as a guideline. Counting the $\delta^{18}$O winter minima in the deeper section of the core suggested 1958 as the date for the bottom of the core (12.93 m w.e.). Using the identified winter minima, we can identify tentative depths for summer maxima (Figure 2a). Most of the horizontal dashed lines in Figure 2a coincide with maxima in the $\delta^{18}$O, indicating a good estimate of the annual cycle

using winter minima and SMB 2007–2011 as a reference.

Furthermore, volcanic horizons are used to corroborate the dating and estimate the dating uncertainty. We use maxima (values above the mean + $2\sigma$ level) in the non-sea salt sulphate (nssSO$_4^{2-}$) concentrations of the KC core to identify volcanic horizons (Figure 4). NssSO$_4^{2-}$ was calculated from the mean seawater composition using Na$^+$ as standard ion, using the equation:

$$\left[\text{nssSO}_4^{2-}\right] = \left[\text{SO}_4^{2-}\right]_{total} - k \times \left[\text{Na}^+\right]_{total} \qquad\qquad \text{Equation 1}$$

With $k = \frac{\left[\text{SO}_4^{2-}\right]_{sea\,water}}{\left[\text{Na}^+\right]_{sea\,water}} = 0.06$, when ion concentrations are in μmol L$^{-1}$. Peaks in nssSO$_4^{2-}$ are assigned to known volcanic eruptions that could alter snow composition at the drilling site, using the Volcanic Explosivity Index (VEI) (Global Volcanism Program, Smithsonian National Museum of Natural History, http://www.volcano.si.edu/). The VEI is a relative measure of the explosiveness of a volcanic eruption based on the volume of released material, plume height, and qualitative

remarks (e.g. by defining an eruption from *gentle* to *mega-colossal*); these parameters are used to construct an open-ended logarithmic scale starting with VEI = 0. Only eruptions with VEI ≥ 3 were considered in this analysis (Table 3). We attribute the nssSO$_4^{2-}$ peaks in the KC core at depths of 3.8 and 4 m w.e. to the Pinatubo volcanic eruption (1991). These depths correspond to the years 1994 and 1993 from the annual layer counting timescale, but delays of 1-2 years between eruption and deposition are commonly observed; the Pinatubo signal has been reported in the 1993 Antarctic snow layer (Cole-Dai et

al., 1997). Another nssSO$_4^{2-}$ peak found at 6.41 m w.e. could originate from the El Chichón volcanic eruption (1982), at a depth corresponding to the year 1983 based on the annual layer counting. Both volcanic horizons have been previously identified in dielectric profiles of other cores drilled in the region (Schlosser et al., 2012 and 2014). Additional volcanic eruptions potentially observed in the nssSO$_4^{2-}$ record and SMB between potential volcanic horizons are listed in Table 3. The error in the dating by annual layer counting is conservatively estimated to be ± 3 years, based on the maximum difference

between the Pinatubo volcanic signals found in the nssSO$_4^{2-}$ record (i.e. peak 1a, Table 3) and the eruption date.





## 4.2 Surface mass balance

Annual SMB in the cores were calculated from distances between summer maxima in the $\delta^{18}O$ record (Figure 5 and Table 1). The average annual SMB for the full period covered by the KC, KM, and BI cores is estimated to be 0.24, 0.68 and 0.70 m w.e., and the average SMB for the common period covered by all three cores (1996–2012), is 0.21, 0.70 and 0.71

5   m w.e., respectively. The lowest inferred annual SMB values at KC, KM and BI were 0.11 m w.e. yr$^{-1}$ (1986), 0.39 m w.e. yr$^{-1}$ (2005) and 0.40 m w.e. yr$^{-1}$ (2004), respectively, while the highest values were 0.45 m w.e. yr$^{-1}$ (1982), 0.95 m w.e. yr$^{-1}$ (2011) and 1.21 m w.e. yr$^{-1}$ (2011), respectively.

These SMB values generally agree with other estimates at FIS (Sinisalo et al., 2013; Schlosser et al., 2014), obtained from shallow cores (1983–2009) and stakes (2010–2012). Furthermore, the anomalous high snowfall in DML during 2009 and

2011 recorded by GRACE satellite data (Boening et al., 2012), and by stake data at FIS (Sinisalo et al., 2013) appear to be reflected in the SMB records of KM and BI (KM: 0.78 and 0.95 m w.e.; BI: 1.00 and 1.21 m w.e., in 2009 and 2011, respectively) (Figure 5).

SMB derived from stake measurements in the vicinity of the core sites at the three ice rises in 2013 are similar to average SMB values from cores at KC (0.22 and 0.24 m w.e. yr$^{-1}$ from the stake and core data, respectively) and BI (0.73 and

0.70  m w.e. yr$^{-1}$), but differ at KM (0.38 versus 0.68 m w.e. yr$^{-1}$). Differences in point estimates for single years are to be expected given the spatial variability of snow accumulation. The spatial variability of SMB on the ice rises from stake and GPR data will be presented elsewhere.

In all three cores, there are ice layers of varying thickness, indicating that some melt occurs in some summers; we have no evidence, however, for mass transport between annual layers. There is no direct correspondence between SMB, $\delta^{18}O$ and the

ice layers in the core from KC (Figure 6), as it has been shown previously from the core S100 (Kaczmarska et al., 2006). It is likely that the development of ice lenses is a local process that depends on several factors, including air and snow pack temperatures.

In general, SMB at the two coastal sites KM and BI is higher compared to the more inland KC site, in agreement with other studies in Antarctica (Sinisalo et al., 2013). In addition, the topography of the individual ice rises is another key determining

factor. While KM and BI are relatively symmetrical domes, KC is more elongated, with a ridge axis stretching from SW to NE (Figure 1). Therefore, air transported from the NNE during a precipitation event is lifted over a longer and gentler slope at KC than at KM and BI, which can lead to a higher loss of moisture on the windward side. Wind-redistribution is critical to accumulation patterns. Networks of stake measurements across KC and KM show an uneven snow distribution, with three-fold higher accumulation on the lowest-elevation upwind side, compared to the summit (Lenaerts et al., 2014). This spatial

pattern is well replicated with the regional atmospheric climate model RACMO2, although an accurate DEM is critical in such comparisons.

Previous studies from coastal sites in the same area of DML have reported large temporal and spatial SMB variability (Melvold, 1999; Kaczmarska et al., 2004; Schlosser et al., 2014). The SMB records from the two coastal cores KM and BI





reveal high interannual variability and no significant long-term trend during the period 1995(96)–2014. At the more inland KC site, SMB variability is lower, but there is also a weak, yet significant (at 95 % confidence level), negative SMB trend of -0.002 m w.e. yr$^{-1}$ for the period 1958–2012. This is in agreement with previous SMB studies done in the whole region of western DML (Isaksson and Melvold, 2002; Kaczmarska et al., 2004; Schlosser et al., 2014), which also document a

5 negative trend in SMB during the 20$^{th}$ century. A comparison between annual SMB calculated in the cores taken at KC, KM, BI, S100 (Kaczmarska et al., 2004), and a composite core constructed averaging the annual SMB from four firn cores (M2, G3, G4 and G5) retrieved at Trolltunga and Jutulstraumen (Schlosser et al., 2014) (Table 1), is shown in Figure 5. Overall, SMB values from the KM and BI cores are higher than for the KC, S100 cores and the composite record. Furthermore, the decreasing trend at KC agrees well with that found in the S100 and in the FIS composite core. One proposed mechanism for

decreasing SMB since the 1980s is the occurrence of stronger zonal wind flow with lower-amplitude long waves during the same period, flow characterized by the Southern Annular Mode (SAM) index (Schlosser et al., 2014). During high SAM index phases meridional moisture flux decreases and consequently, there is less precipitation and SMB along the coast (Schlosser et al., 2014).

### 4.3 Water stable isotopes

Figure 7 shows the raw and annual $\delta^{18}$O, $\delta^2$H and $d$ (deuterium excess) data for the KC, KM, and BI sites. The higher accumulation rates at KM and BI allow high-resolution water stable isotope records (average 20 data points per year), whereas resolution at KC is much lower (average 7 data points per year). Table 1 shows the median values of $\delta^{18}$O, $\delta^2$H and $d$ of the KC, KM, and BI cores. The $\delta^{18}$O and $\delta^2$H signals at KM and BI (Figure 7b and c) show pronounced seasonality, with seasonal amplitudes up to 10 ‰ and 78 ‰, for $\delta^{18}$O and $\delta^2$H, and up to 10 ‰ for $d$. Stable isotope ratios for KM and BI

are similar, while KC has generally lower values of $\delta^{18}$O and $\delta^2$H than the two coastal cores. The core site at BI is 130 m higher than that of KC and KM (see Table 1), such that differences in isotope ratios can be attributed to local effects. Similar results were found by Fernandoy et al. (2010) for firn cores drilled in the hinterland of Neumayer Station, where higher-elevation cores did not always have lower $\delta^{18}$O values.

The present study is among the few that also includes deuterium, and the first involving such data from FIS. It is difficult to

25 reliably determine the seasonal cycle of $d$ for the analysed cores; we cannot date the cores accurately at the sub-annual level, post-depositional processes such as water vapour diffusion in the firn column may alter the $d$ profile, and finally the $d$ time-series are relatively short. Nevertheless, we use the derived age-models to estimate the intra-annual variability in $d$ for the two higher-resolution cores at KM and BI. The results (Figure 8) suggest agreement with known $d$ seasonal curves for the high south, with a maximum (minimum) for austral autumns (summers) (e.g. Delmotte et al., 2000; Pfahl and Sodemann,

2014), thereby lagging by 3–4 months the corresponding seasonal peaks in $\delta^{18}$O (Figure 8). We note that because of the method we use to construct the core chronologies, the seasonal curve of $\delta^{18}$O is fit to have a maximum (minimum) in the



first (sixth) month of the year. However, the good agreement with observational data from other locations supports the phase shift we infer.

There are various potential climate controls on $d$ in precipitation, with sea surface temperature and/or relative humidity at moisture origin being considered to provide major influences. Recent studies show that the relationship of $d$ and moisture source parameters is more complex than previously thought (Steen-Larsen et al., 2014; Dittmann et al, 2016).

High correlation coefficients between the annual $d$ and austral spring to summer SAM indices of -0.55 (significant at the 95 % confidence level) for KM and -0.33 (not significant) for BI cores point to a possible role of seasonal changes in moisture transport and precipitation in the area in shaping the annual isotopic signal. A positive SAM index is generally associated with stronger zonal westerlies and comparatively little exchange of moisture and energy between middle and high latitudes (Marshall et al., 2013; Schlosser et al., 2016), hence increasing the contribution of local, less depleted of $\delta^{18}$O, moisture sources in precipitation (Noone and Simmonds, 2002). Decreased meridional southward moisture transport during the positive SAM phase may vary the annual moisture balance towards a higher fraction of local spring and summer moisture. Compared to moisture originating from more remote lower latitude sources, local sources in spring and summer typically have lower $d$ values, leading to generally negative annual $d$ anomalies preserved in the snow. The multidecadal positive trend in SAM (e.g. Marshall 2003), which is especially pronounced for austral summers, may in turn drive the weak negative trend of 0.1 ‰ per decade (not significant at the 95 % confidence level) in $d$ detected in the longer KC core, also contributing to an observed positive trend in $\delta^{18}$O in the regional core network.

Figure 9 shows the mean annual $\delta^{18}$O for the KC, KM and BI cores compared to the S100 and composite core from Schlosser et al. (2014). Overall, annual $\delta^{18}$O values at the three ice rise cores are higher than at S100 or for the composite core. However, both the inferred multiannual means and the standard deviations of $\delta^{18}$O for the three ice rise cores fall within the typical range of variability for other cores from the coastal DML (Altnau et al., 2015). The positive linear trends in $\delta^{18}$O observed in the KC and BI cores also agree well with linear trends reported for the S100 and FIS composite cores (Kaczmarska et al., 2004; Schlosser et al., 2014). However, none of the linear trends observed in the KC, KM or BI cores are significant at the 95% confidence level. Similar to earlier studies (e.g. Schlosser et al., 2014) no correlation is found between $\delta^{18}$O of the ice rise cores and measured air temperature at Neumayer Station, the closest station suitable for comparison. Neumayer is situated on a small ice shelf, with synoptic conditions similar to FIS; no temporal trend is found in air temperature since the founding of the station in 1981. Likewise, no relationship between stable isotopes and SMB is seen in the ice rise cores (compare with Figure 5). This confirms previous studies, which find poor correspondence between SMB and proxy-temperatures, suggesting that it is large-scale atmospheric circulation rather than the thermodynamic relationship between SMB and temperature that is the determining factor here. This was also found in a recent study by Fudge et al. (2016), who investigated the temperature-SMB relationship using data from the WAIS Divide ice core.

Due to the restricted length of the KM and BI cores, further analysis of the spatial and temporal differences of water stable isotopes at the three ice rises in a climatic context would be speculative. However, the ice rises coring sites show potential



for investigating past variations in water stable isotopes given on the well preserved profiles, with annual to bi-annual resolution at the KC site, and subannual resolution at the KM and BI sites.

## 5 Discussion and conclusions

Hitherto, small ice rises in Antarctica have not been fully utilised as ice core sites. Based on the data presented we conclude that the stratigraphic records of water stable isotopes and major ions (in particular MSA and $Na^+$) are well preserved during the last decades so that reliable annual dating can be performed. Neither the stratigraphy nor the chemistry profiles in the cores suggest that there is substantial surface melting or percolation at the sites, which would perturb the stratigraphic signal. This can be further seen by the number of ice lenses, ice lenses thickness, and density profiles available for the KC site (Figure 6) which shows that most ice lenses are thinner than 1 cm, with the thickest having 1.5 cm. In terms of ice content per meter of firn, KC core has in average no more than 3 % of ice per meter during the period 1958–2012. Considering the above, the core timescales were constructed based in annual layer counting of $\delta^{18}O$ (KC, KM and BI) together with the identification of volcanic layers using the $nssSO_4^{2-}$ record (KC). These approaches appear to provide reliable methods for the dating of these firn cores involving dating errors of ±1 year (KM and BI) and ±3 years (KC).

The well preserved physico-chemical properties of the cores means that SMB and information related to both short- and long-term regional variability of ion sources and moisture, can potentially be reconstructed over greater time scales if longer cores were to be retrieved from these locations.

The SMB records from the different sites show that topography likely leads to local effects that are superimposed on the regional climate signal. This is particularly the case for the coastal KM and BI ice rise sites which have much higher SMB (and hence higher variability) than the KC, S100 and composite cores, with trends also opposing the findings from the other records. It is therefore of great importance to further investigate if the data from the KM and BI ice rises have also a regional significance.

The longest SMB record, from KC, is in general agreement with other regional ice core records (S100 and the composite core), and shows that the negative trend observed during the 20[th] century in the longer S100 core retrieved nearby (Figure 1) continues during the first decade of the 21[st] century. This decrease in SMB since the 1980s has been proposed to be related to diminishing meridional moisture flux and consequently, a decrease in precipitation and SMB at FIS (Schlosser et al., 2014). This decrease in SMB is, however, not seen in the coastal KM and BI cores, which show an increase in SMB, although the trends are not statistically significant. Since SMB is highly variable in both time and space, these differences in annual SMB and trends are not surprising and could easily be the result of factors such as differences in distance to the ice shelf edge and/or elevation. Both KM and BI cores are located facing the ice shelf edge and at more than 200 m a.s.l., whereas the S100 core, although also located at the ice shelf edge, is closer to sea level (48 m a.s.l.). KC is located at an elevation of more than 200 m a.s.l. but is completely surrounded by the ice shelf, which, together with the topographical and post-depositional effects presented in section 4.2, may lead to the reduced annual snow accumulation there, compared to the other ice rises.



The most commonly used Antarctic SMB maps (e.g. Arthern et al. 2006; Monaghan et al., 2006; Lenaerts et al., 2012) all have a too low resolution to properly incorporate the high variability that ice rises induces.

The first data available for $d$ at FIS point to a possible role of seasonal moisture transport changes and precipitation in shaping the annual isotopic signal at the area, as inferred from the high correlation found between annual $d$ in the KM and BI cores and austral spring to summer SAM indices. When considering the factors behind the water stable isotope values, the poor correspondence between SMB and proxy-temperature derived from water stable isotopes suggests that large-scale atmospheric circulation patterns are the determining factors for isotope ratios, in agreement with previous studies at FIS (Schlosser et al., 2014).

In summary, the ice rises are suitable drilling sites for the retrieval of longer cores if local influences are kept in mind when reconstructing the past climate and environmental signals recorded in the cores. Especially attractive to retrieve high-resolution (i.e. subannual timescales) ice core records are the KM and BI sites due to their high accumulation rates and well preserved physical and chemical properties, bearing in mind that these sites may also be strongly affected by local snow deposition patterns. On the other hand, the KC location could be considered as the most representative for the climate of the area although, even if it is not possible to obtain subannual dating due to the lower annual snow accumulation at that site. Since drifting snow processes are of major importance on ice rises, detailed knowledge of both topography and the spatial pattern of SMB are required for deciding on possible future ice core locations. Consequently, the three ice rises investigated here offer attractive locations for the retrieval of longer ice cores that would contribute to elucidate the climate and environmental history of the FIS, and to infer its role in a changing climate.

### Acknowledgements

We are grateful to the number of people who helped to collect, transport, sample and analyse the firn cores and snow pits at FIS. The financial support came from the Norwegian Research Council through NARE and the Centre for Ice, Climate and Ecosystems (ICE) at the Norwegian Polar Institute in Tromsø, Norway.

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





**Table 1:** Ice rises core locations, sampling details, SMB rates derived from the KC, KM and BI $\delta^{18}O$ records as annual average values between summer maxima. Median water stable isotope deltas (in ‰) quantified in each core are also shown. Distance of the core locations to the ice shelf edge was obtained using the GIS package *Quantarctica* (www.quantarctica.org). Both annual layer counting and volcanic horizons identified in the $nssSO_4^{2-}$ were used to obtain timescales for the cores. Significant values (at 95 % confidence level) are shown in bold. (*) refers to Schlosser et al. (2014).

| Site | Location | Elevation | Core length (Ice depth) | Shortest distance from the ice shelf edge | Time coverage (Dating error) | Average SMB rate (min., max.) | Slope of the linear regression ($\sigma$) | Median $\delta^{18}O$ $\delta^2H$ $d$ | Slope of the linear regression of $\delta^{18}O$ ($\sigma$) |
|---|---|---|---|---|---|---|---|---|---|
| | | m a.s.l. | m | km | years | m w.e. yr$^{-1}$ | m w.e. yr$^{-1}$ | ‰ | ‰ yr$^{-1}$ |
| KC | 70°31′S, 2°57′E | 264 | 20.0 (460) | 42 | 1958–2012 ($\pm$3) | 0.24 (0.11, 0.45) | **-0.002** ($\pm7\times10^{-4}$) | -19.4 -150.2 4.8 | -0.004 ($\pm$0.01) |
| KM | 70°8′S, 1°12′E | 268 | 19.6 (410) | 12 | 1995–2014 ($\pm$1) | 0.68 (0.39, 0.95) | 0.004 ($\pm9\times10^{-3}$) | -17.5 -133.6 5.9 | 0.03 ($\pm$0.05) |
| BI | 70°24′S, 3°2′W | 394 | 19.5 (460) | 10 | 1996–2014 ($\pm$1) | 0.70 (0.40, 1.21) | 0.006 ($\pm1\times10^{-2}$) | -17.6 -134.5 6.3 | 0.03 ($\pm$0.05) |
| Composite core* (M2, G3, G4, G5) | | | | | 1983–2009 | 0.38 (0.15, 0.57) | **-0.007** ($\pm2\times10^{-3}$) | - - - | **0.06** ($\pm$0.02) |
| **Overlapping period 1996-2012** | | | | | | | | | |
| KC | | | | | | 0.21 (0.42, 0.11) | -0.007 ($\pm5\times10^{-3}$) | -19.1 - - | **-0.27** ($\pm$0.09) |
| KM | | | | | | 0.70 (0.39, 0.95) | 0.01 ($\pm$0.01) | -17.4 - - | 0.01 ($\pm$0.04) |
| BI | | | | | | 0.71 (1.21, 0.40) | 0.01 ($\pm$0.01) | -17.2 - - | 0.03 ($\pm$0.05) |





**Table 2:** Median ion concentrations (in μmol L$^{-1}$) in the KC, KM and BI firn cores. Ion concentrations at the top 2 m of the KC, KM and BI cores were not measured. Non-detected concentrations were set as half the detection limit for each ion.

| Median | Period (Year) | MSA | SO$_4^{2-}$ | Na$^+$ |
|--------|---------------|-----|-------------|--------|
| | | μmol L$^{-1}$ | | |
| KC | 1958–2007 | 0.2 | 1.8 | 9.4 |
| KM | 1995–2014 | 0.3 | 4.5 | 57.7 |
| BI | 1996–2014 | 0.4 | 1.9 | 19.0 |

**Table 3:** Volcanic eruptions inferred from the KC nssSO$_4^{2-}$ concentrations. Only volcanoes with a volcanic explosivity index ≥ 3 were considered. The SMB rate was obtained using the timescale obtained by $\delta^{18}$O cycles counting. Ref.: [1]Karlöf et al. (2000), [2]Palmer et al. (2001), [3]Nishio et al. (2002), [4]Stenni et al. (2002), [5]Zhang et al. (2002), [6]Kohno and Fujii (2002), [7]Global Volcanism Program, and [8]Vega (2008).

| Peak No. | Bottom depth m w.e | Year in the cycles counting timescale | Assigned volcano (Year of eruption)⁻ | SMB rate m w.e yr$^{-1}$ (Period) | Ref. |
|----------|--------------------|---------------------------------------|--------------------------------------|-----------------------------------|------|
| 1 | 4.00 | 1992.8 | Pinatubo, Philippines (1991) | 0.21 (1991–2011) | 1, 2, 3, 4, 5 |
| 1a | 3.80 | 1993.9 | Pinatubo, Philippines (1991) | 0.21 (1991–2011) | 1, 2, 3, 4, 5 |
| 2 | 6.41 | 1982.8 | El Chichón, Mexico (1982) | 0.26 (1982–1990) | 6 |
| 3 | 9.89 | 1970.3 | Deception island, Antarctic Peninsula (1970) | 0.26 (1970–1981) | 7 |
| 4 | 10.20 | 1968.3 | Deception island, Antarctic Peninsula (1967) | 0.16 (1967–1969) | 8 |
| 5 | 11.91 | 1961.7 | Puyehue, Cordón Caulle, Chile (1960) | 0.28 (1960–1966) | 8 |
| 6 | 12.64 | 1959.1 | Carran-Los Venados, Chile (1955) | 0.33 (1959) | 3, 8 |

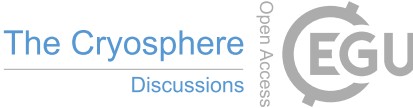



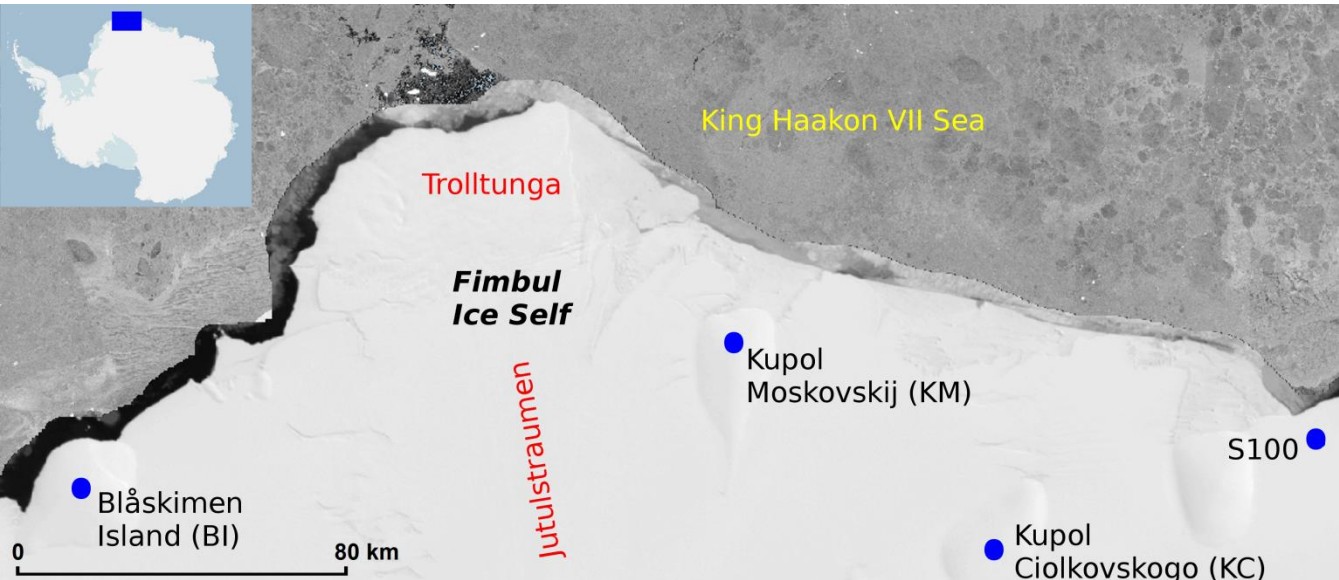

**Figure 1:** Satellite image of the Fimbul Ice Shelf (FIS), East Antarctica. The KC, KM, BI and S100 core sites are shown together. Map image is from the Radarsat Antarctica mosaic available in the GIS package Quantarctica (www.quantarctica.org). Information regarding additional sampling sites and traverses in FIS can be found in Schlosser et al. (2014).



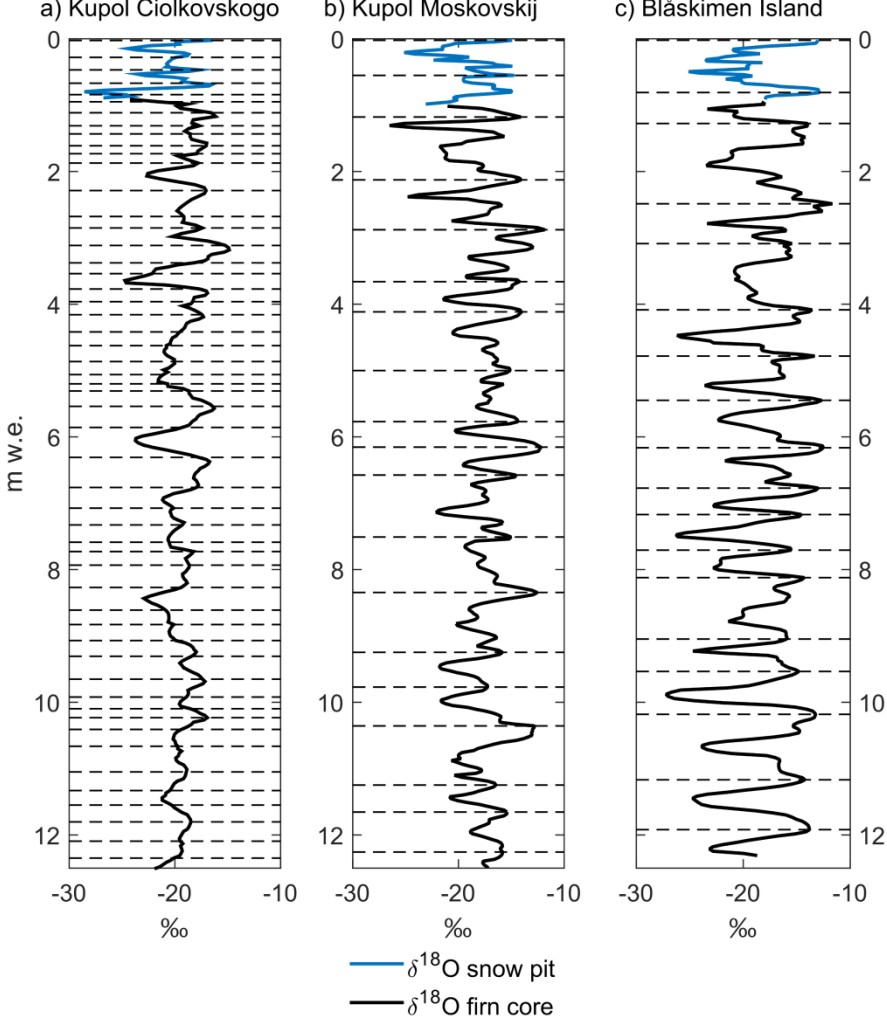

**Figure 2:** $\delta^{18}$O–depth profiles of a) KC, b) KM, and c) BI. The blue lines indicate the $\delta^{18}$O values for the 2-m snow pits at each core site. Seasonal variations are used to date the KM and BI cores; horizontal lines mark the summer maxima inferred in the KC core and identified in the KM and BI cores.





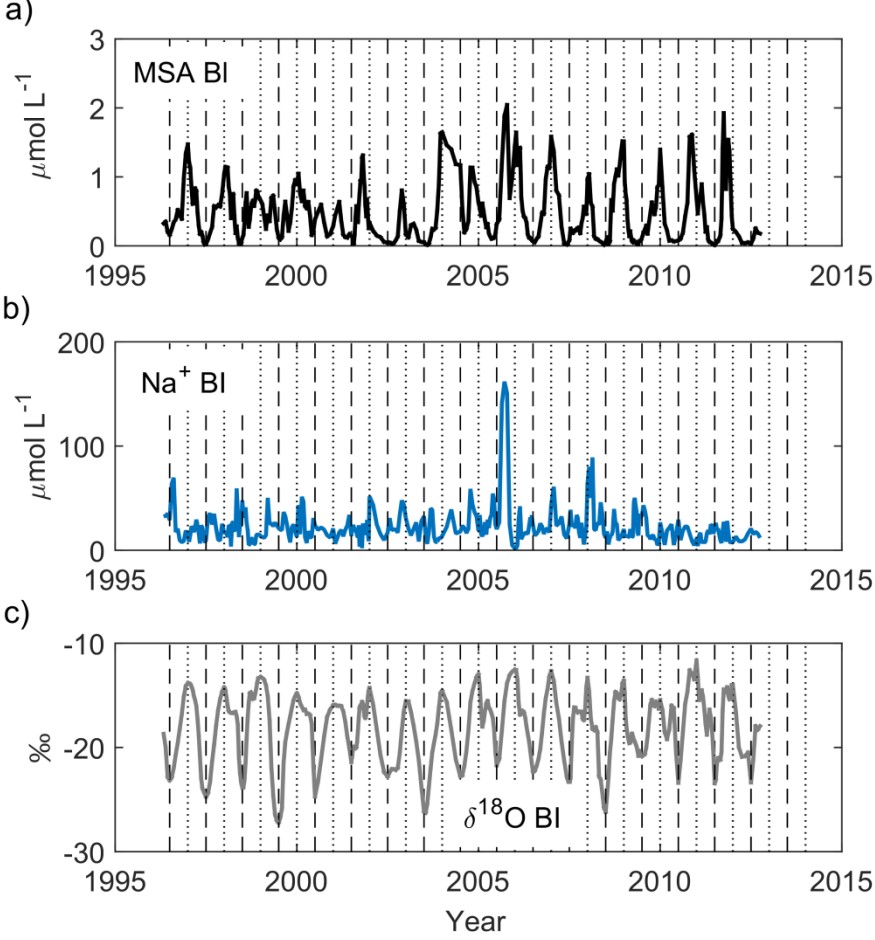

**Figure 3:** Seasonality of a) MSA, b) Na$^+$ and c) $\delta^{18}$O for the BI core. Dashed lines and dotted lines indicate winter (summer) minima (maxima).





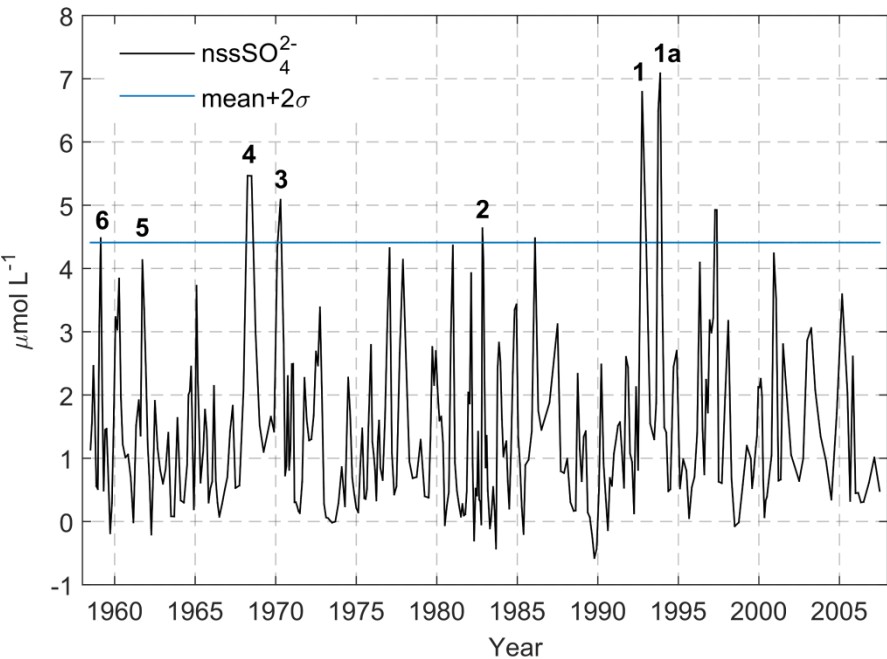

**Figure 4:** Non-sea salt $SO_4^{2-}$ concentrations measured in KC, using the timescale derived from annual layer counting in the $\delta^{18}O$ profile. Potential layers of volcanic eruptions are marked with numbers: 1, 1a (Pinatubo), 2 (El Chichon), 3, 4 (Deception Island), 5 (Puyehue, Chile), 6 (Caran-Los Vernados, Chile) and summarized in Table 3.



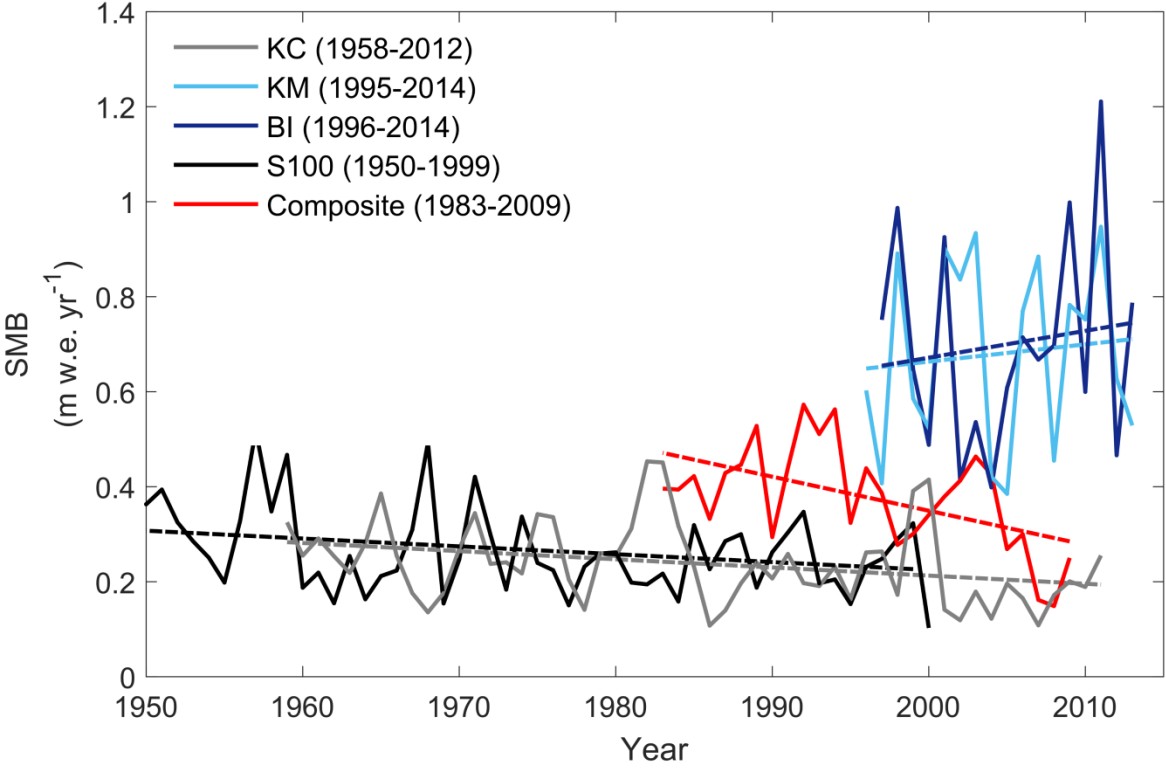

**Figure 5:** Annual SMB for KC, KM, and BI cores compared to S100 (Kaczmarska et al., 2004) and the composite FIS core record (Schlosser et al., 2014). The dashed lines are the linear regression for the entire period covered by the respective cores.





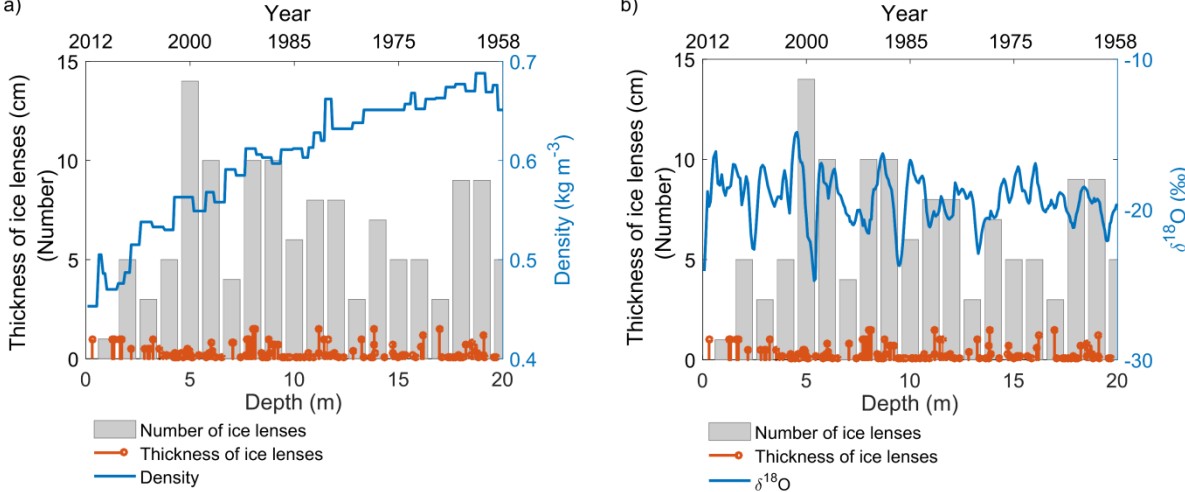

**Figure 6:** a) Number of ice lenses per meter (bars), ice lenses thickness (stems) and density profiles (line) available at KC. b) Same as a) but with the $\delta^{18}O$ profile (line) instead of density.





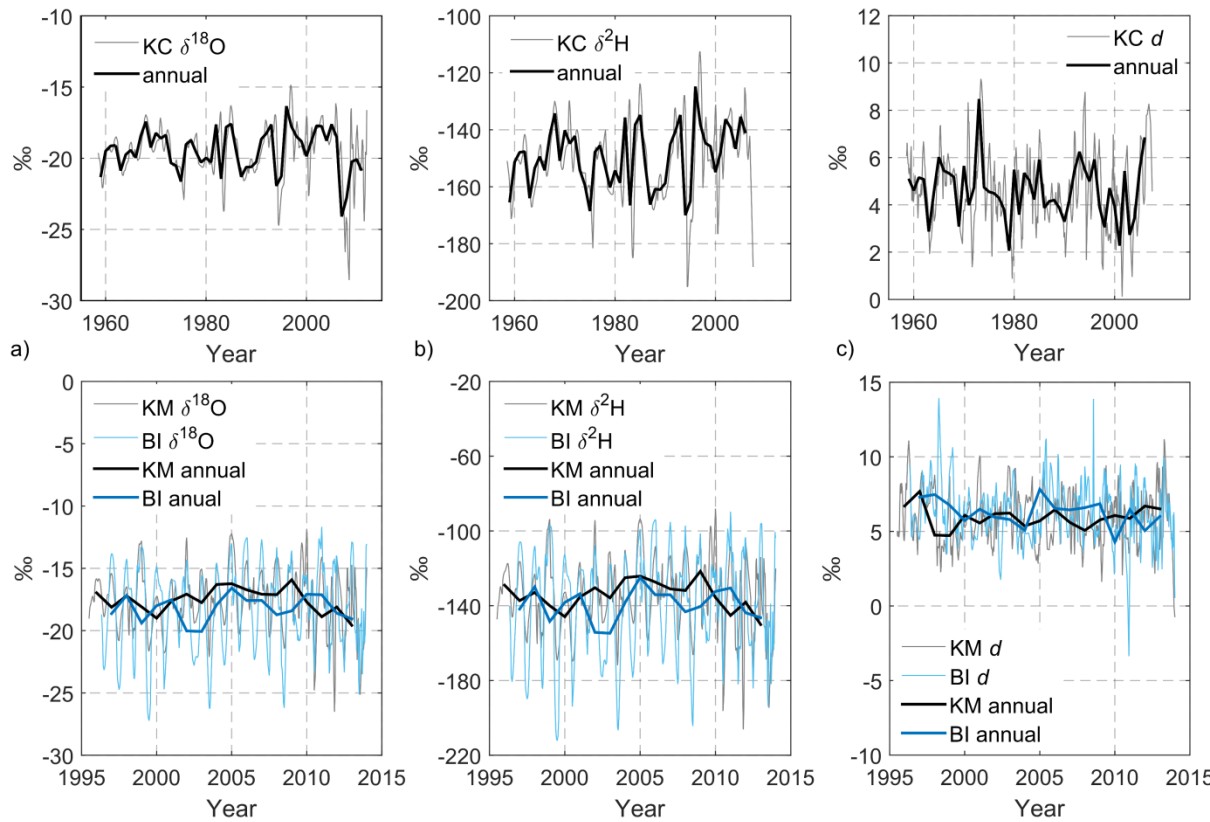

**Figure 7:** Water stable isotope data for KC, KM, and BI: a) $\delta^{18}$O, b) $\delta^{2}$H, c) deuterium excess (*d*).





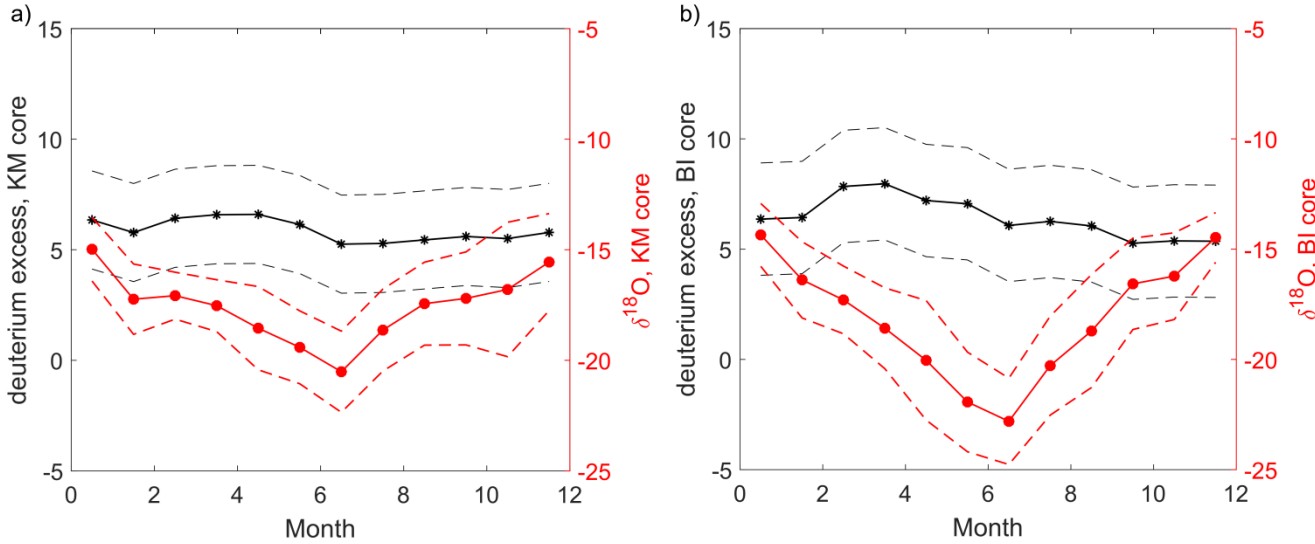

**Figure 8:** Seasonal variations in $d$ (black) and $\delta^{18}O$ (red) in cores a) KM and b) BI. Dashed lines show $\pm\ \sigma$.





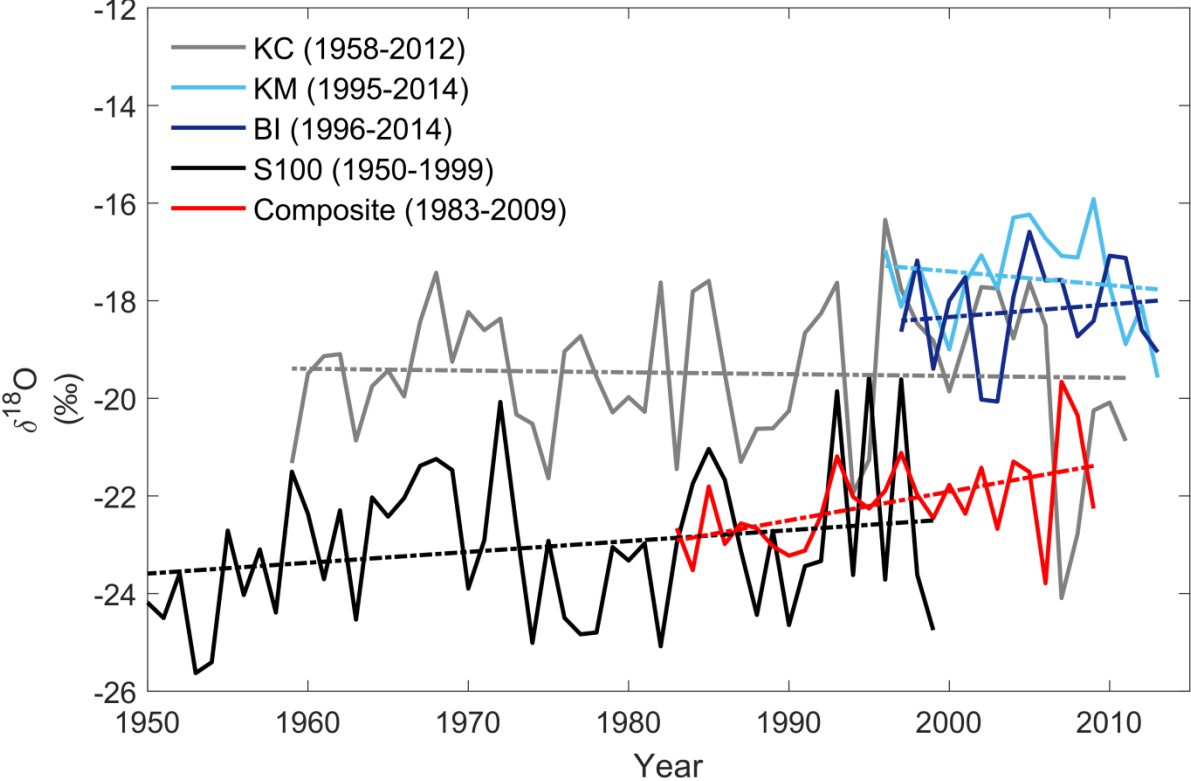

**Figure 9:** Mean annual $\delta^{18}O$ for KC, KM, and BI compared to S100 (Divine et al., 2009) and the composite FIS core record (Schlosser et al., 2014). The dashed lines are the linear regression for the entire period covered by the respective cores.