# Peer review of "Surface mass balance and water stable isotopes derived from firn cores on three ice rises, Fimbul Ice Shelf, Antarctica"

_The Cryosphere, 2016_

## Referee Comment (RC1) · E. Thomas (Referee) · 2 Aug 2016

The paper presents surface mass balance data for the Fimbul ice shelf based on three firn cores. The paper is clearly written and well presented. The authors provide a good introduction to the region and the previous studies that have taken place. The paper should be accepted and I only have very minor comments to make.

KC core- I am not convinced of the dating for the KC core. The seasonal cycles in figure 2 are very difficult to see. The use of nssSO4 tie points should help but the volcanic horizons presented in figure 4 could easily be shifted. The disadvantage of the coastal locations are the increased variability in the nssSO4 record and very few large volcanic events during the time period investigated. Future drilling at this site

would need alternative dating methods in place.

Page 5, line 1 – the words "ice rises cores" is a bit tricky to read. Could you perhaps rephrase to "ice rises and drill sites" ?

Page 5, Line 27 – You make the assumption that there is uniform precipitation throughout the year. Is this true? Have there been studies on the seasonality of precipitation in this region you could refer to? If not, what does the reanalysis data suggest for the seasonality of precipitation on FIS?

Page 7, Line 24 – use of the words "In addition" is not necessary

Page 11, Line 10 sentence starting "especially attractive..." needs rewording. Suggest "The KM and BI sites are attractive ....

Figure 2 – the seasonal cycle at KC is very difficult to distinguish. Does altering the x-axis make the peaks any clearer?

---

## Referee Comment (RC2) · M. Frezotti (Referee) · 31 Aug 2016

This paper presents the surface mass balance and water stable isotopes reconstructed using ice cores on three ice rises, Fimbul Ice Shelf, Antarctica. The main tools are high resolution oxygen isotope and identification of volcanic horizons using nssSO4 data. The paper contributes to the reconstruction of the present reconstruction of climate in the coastal area of DML area.

The manuscript subject is very appropriate for "The Cryosphere" and it is well written, but the analysis and presentation of the data and interpretation must be improved.

In the manuscript the Authors point out that at ice rises is present asymmetrical local

meteorology conditions and surface mass balance respect to surrounding ice shelf, but the Authors do not report any information about the site location of core site respect to the ice rise and climatic condition. The location of core site respect to ice rises morphology is very important parameters for the interpretation of spatial and temporal variability of SMB and isotope.

The dating of core KM and BI is well constrained by isotope and MSA signal (seasonality of MSA for KM must be reported in a figure similar to 3, Na do not provide information and could be removed), whereas the stratigraphy of KC must be improved using seasonal signal of nssSO4, MSA and isotope, the volcanic signal reported is not well constrained respect other nssSO4 antarctic ice core stratigraphy: Double peaks of Pinatubo (melting layer, between? Cerro Hudson eruption?) Agung eruption of 1963 is one of the signals reported in Antarctic ice cores, and it is not observed instead of two others eruptions (Deception and Puyehue), could authors provide a comment?

BI and KM present similar isotope value (with 130 m of elevation difference) and presenting less depleted isotope respect to other ice cores located on ice shelf that authors correctly attributed to reduction of inversion layer, KC core present depleted isotope value and much lower SMB, the position of KC does not appear on the ice rises on the base of the data. The SMB and isotope value appear strictly correlated to morphological position, the discussion of the result must be introduced by the analysis of climate and morphological analysis of core site also of S100 and M2, G3, G4 and G5 site (information of these sites must be reported on Figure 1 and table 1, and if are available the firn temp a -10 m), and the profile of cores (M2, G3, G4 and G5) must be reported in figure 5 and 9.

SMB of KC, S100 and Composite appear similar, whereas the KM and BI are similar, but significant different, clearly the two groups represent different climate condition and history, authors must provide some explanation in the discussion. Clearly ice rises present different SMB and isotope, the KC SMB is "medium/high" for Antarctic standard condition, and with this value normally the seasonal signal is well preserved (see

WAIS), the absence of annual signal is related at others factor that must be analysed.

KC isotope profiles $\partial$O18 and deuterium appear on figure 7 very differents, could be due to the melting? KC profile appears smooth in the lower part of profile, could the Authors comments?

At pag 7 line 15 it reports stake measurements: how many per each site? Single stake respect to single core? Explain

Pag 7 line line 18, on the base of figure 6 the melting occurs several time per year, not "some summers". The importance of melting layer in the stratigraphy must be analysed in more depth with comparison of position respect to nssSO4, isotope and MSA, in particular for KC cores. The site present a "relatively high" SMB (0.24 m we yr) and it is anomalous that the seasonal stratigraphy is not preserved. Could be the combination of windscouring and melting obliterated the seasonal signal?

Pag 9 if the data are not significant at 95% and it is not reported in figure 7, I do not understand their importance.

Pag 10 line 32, the topographical and post-depositional effects are not sufficient presented in section 4.2

The different trend in SMB between ice shelf and ice rises must be also analysed respect the observation of the behaviour of SMB Antarctica proposed by Frezzotti et al., 2013.

In the figure 1 should reported the contour line at 10 m.

---

## Author Comment (AC1) · 8 Oct 2016

Final author comments on "Surface mass balance and water stable isotopes derived from firn cores on three ice rises, Fimbul Ice Shelf, Antarctica" by C. P. Vega et al.

To the referees:

The authors would like to thank the referees for the time taken to review the manuscript. We truly value the general and specific comments and suggestions made by the referees which have been very helpful when revising the manuscript. We agree in most of the comments made by the referees, and we have included their valuable suggestions in the revised version as long as it was possible. In addition to the referees'

suggestions, we have also extended the discussion on d in section 4.3 in light of recent literature that was not included in the previous version of this manuscript. Each of our responses has been noted as CV. following each of the referee's comments.

Response to referee #1, E. Thomas: Ref. #1: The paper presents surface mass balance data for the Fimbul ice shelf based on three firn cores. The paper is clearly written and well presented. The authors provide a good introduction to the region and the previous studies that have taken place. The paper should be accepted and I only have very minor comments to make.

1. KC core- I am not convinced of the dating for the KC core. The seasonal cycles in figure 2 are very difficult to see. The use of nssSO4 tie points should help but the volcanic horizons presented in figure 4 could easily be shifted. The disadvantage of the coastal locations are the increased variability in the nssSO4 record and very few large volcanic events during the time period investigated. Future drilling at this site would need alternative dating methods in place.

CV. We completely understand Ref. #1 point. In fact, it was the lack of a good visualization of the seasonal cycles in the KC core that directed us to use the mean SMB for the period 2007–2011 obtained for the KC snowpit (in which the amplitude of the seasonal cycles are slightly better preserved than in the core, and in addition, the inferred SMB can be compared to stake data obtained at the site) as a reference to find possible annual layers in the KC d18O profile. This mean SMB was in agreement with stake values for the KC site, therefore, we considered this reference as sufficiently good to stablish a preliminarily time scale which was later complemented with the use of the nssSO42- peaks as tie points. We agree with Ref. #1 in that the volcanic horizons presented in Table 3 could be shifted and consequently, we estimated the dating error as ±3 years corresponding to the shifting in date based on the maximum difference between the Pinatubo volcanic signals found in the nssSO42- record and the eruption date. In addition, and following Ref. #2 suggestions, we have considered peak 1 in Fig. 4 and Table 4 as to be the Cerro Hudson eruption since the appearance of a distinguishable nssSO42- peak before the Pinatubo eruption has been previously reported by Cole-Dai et al. (1997), and assigned to the Cerro Hudson eruption (Please refer to the answer to Ref. #2, point 8).

We agree with Ref. #1 in that complementing dating methods are needed in order to stablish a more precise time scale for future cores drilled at this site, as for example, counting annual cycles not only in water stable isotope data but also in other chemical species such as MSA and Na+, and also using 3H measurements to identify additional tie points. Since this point is important to emphasize in the manuscript, we have added the following sentence at the end of section 4.1: "The timescale error could be reduced by complementary dating methods, such as annual cycles counting of chemical species (e.g. MSA and Na+), and 3H measurements, for future cores drilled at these sites."

2. Page 5, line 1 – the words "ice rises cores" is a bit tricky to read. Could you perhaps rephrase to "ice rises and drill sites"?

CV. The sentence now says: "Table 1 presents the location of the drill sites, maximum elevation of the ice rises, and recovered core lengths."

3. Page 5, Line 27 – You make the assumption that there is uniform precipitation through- out the year. Is this true? Have there been studies on the seasonality of precipitation in this region you could refer to? If not, what does the reanalysis data suggest for the seasonality of precipitation on FIS?

CV. Extensive records of precipitation at the core sites or at Fimbul Ice Shelf are, to our knowledge, not existent. The assumption of uniform precipitation throughout the year at the core sites was made on the basis of the precipitation regime at Dronning Maud Land (DML) by Schlosser et al. (2008) which showed high temporal variability. In their study, the authors used the high-resolution Antarctic Mesoscale Prediction System (AMPS) archive data to study the precipitation regime of DML and found that high inter-annual variability of the precipitation monthly sums due to the influence of cyclone

activity affecting both coastal and inland regions. Figure 5 in Schlosser el al. (2008) shows that for Neumayer, the closest station to the study sites, two precipitation maxima are identifiable for the period 2001–2006 (April and October) possible explained by the semi-annual oscillation of the circumpolar trough. As to what extent the same precipitation regime is kept at the Ice Rises is a difficult guess, even more when it comes to the estimation of the sub-annual time scale of the cores, considering that the SMB cannot directly be considered as a measure for precipitation at sites where post-depositional processes are most likely present. Considering the above, we decided to assume a linear precipitation regime throughout the year to simplify the sub-annual estimation of the time scale of the three cores, which we consider is a fair assumption since we mostly analysed the data at an annual resolution in this manuscript. In the cases when the data was interpreted at a sub-annual resolution, e.g. when discussing the d value in the KM and BI cores (Fig. 8), we explicitly noted the limitations of the method used to obtain the sub-annual time scale. We will definitely consider a more detailed approach, e.g. by using ERA-interim reanalysis, to infer the precipitation regime in further work in which sub-annual variations could be interesting to explore in more depth.

4. Page 7, Line 24 – use of the words "In addition" is not necessary

CV. The sentence now says: "The topography of the individual ice rises is a key determining factor."

5. Page 11, Line 10 sentence starting "especially attractive…" needs rewording. Suggest "The KM and BI sites are attractive….

CV. The sentence now says: "The KM and BI sites are attractive to retrieve high-resolution (i.e. subannual timescales) ice core records…"

6. Figure 2 – the seasonal cycle at KC is very difficult to distinguish. Does altering the x-axis make the peaks any clearer?

CV. Unfortunately, modifying the x-axis does not allow for a better visualization of the peaks in Fig. 2 a. This is basically because the amplitude of the values near the top of the snow pit is larger in comparison to the deeper parts of the core. Even not considering the values at the snow pit and; and therefore, setting the x-axis range to match the amplitudes of the values measured in the core, did not improved the visualization of the cycles. We discussed more about this topic in our answer Ref. #1, point 1. Consequently, we have not changed the aspect of Figure 2.

Response to referee #2, M. Frezzotti:

Ref. #2: This paper presents the surface mass balance and water stable isotopes reconstructed using ice cores on three ice rises, Fimbul Ice Shelf, Antarctica. The main tools are high resolution oxygen isotope and identification of volcanic horizons using nssSO4 data. The paper contributes to the reconstruction of the present reconstruction of climate in the coastal area of DML area. The manuscript subject is very appropriate for "The Cryosphere" and it is well written, but the analysis and presentation of the data and interpretation must be improved.

7. In the manuscript the Authors point out that at ice rises is present asymmetrical local meteorology conditions and surface mass balance respect to surrounding ice shelf, but the Authors do not report any information about the site location of core site respect to the ice rise and climatic condition. The location of core site respect to ice rises morphology is very important parameters for the interpretation of spatial and temporal variability of SMB and isotope.

CV. The Referee is correct in pointing this out. Consequently, we have now included 50-m contours at the three ice rises which help to visualise the topography at each of the ice rises (Fig. 1), which is also discussed in section 4.2.

8. The dating of core KM and BI is well constrained by isotope and MSA signal (seasonality of MSA for KM must be reported in a figure similar to 3, Na do not provide information and could be removed), . . .

CV. Fig. 3 shows now MSA, Na+ and d18O seasonality in both the BI and KM cores. We decided to keep the Na+ profile as comparison.

. . .whereas the stratigraphy of KC must be improved using seasonal signal of nssSO4, MSA and isotope, . . .

CV. We have now added MSA and nssSO42- in Fig.6 (c-d, respectively). We have also corrected the x-axis in Fig.6 which now considers the fact that the KC core was drilled from the bottom of a 2 m deep snow pit.

. . .the volcanic signal reported is not well constrained respect other nssSO4 Antarctic ice core stratigraphy: Double peaks of Pinatubo (melting layer, between? Cerro Hudson eruption?) Agung eruption of 1963 is one of the signals reported in Antarctic ice cores, and it is not observed instead of two others eruptions (Deception and Puyehue), could authors provide a comment?

CV. The number of ice lenses at the depth of the nssSO42- attributed to Pinatubo is above the average at the core, however, their average thickness is below the average ice lens thickness for the whole KC core. We could speculate that the second peak attributed to Pinatubo (peak 1 in Table 4 and Fig. 4) could be produced by ion relocation due to meltwater percolation but it is more likely that peak 1 corresponds to the Cerro Hudson eruption (Cole-Dai et al. 1997). Consequently, we have now added the Cerro Hudson eruption as a probable eruption associated to peak 1 in Table 4 and mentioned it in section 4.1 of the manuscript. Peaks 1a and 1 agree with previous works that found that Pinatubo and Cerro Hudson eruptions greatly increased the nssSO42- over Antarctica during the period 1991–1993 (Dibb and Whitlow, 1996; Cole-Dai et al., 1997).

Regarding the Agung eruption in 1963, Ref. #2 is completely right in pointing this out. We realized that the Agung volcano was missing in Table 4 what it should have been and error when editing the final version of the manuscript, and we apologize for this. Consequently, we now listed the Agung eruption together with the eruption of the

[Figure]

Puyehue volcano as possible sources of peak 5. Local glaciological conditions could be the reason of the relatively low nssSO42- peak assigned to Agung in comparison to other sites where the Agung peak shows intensities similar to the Pinatubo peak. Regarding the peaks assigned to the eruptions of Deception Island and the Puyehue volcano, these peaks in nssSO42- have been previously reported in ice cores from James Ross Island, Antarctic Peninsula (Vega, 2008).

9. BI and KM present similar isotope value (with 130 m of elevation difference) and presenting less depleted isotope respect to other ice cores located on ice shelf that authors correctly attributed to reduction of inversion layer, KC core present depleted isotope value and much lower SMB, the position of KC does not appear on the ice rises on the base of the data.

CV. The location of the cores sites is shown in Fig.1 and also the 50-m contours have been added now.

. . .The SMB and isotope value appear strictly correlated to morphological position, the discussion of the result must be introduced by the analysis of climate and morphological analysis of core site also of S100 and M2, G3, G4 and G site (information of these sites must be reported on Figure 1 and table 1, and if are available the firn temp a -10 m), and the profile of cores (M2, G3, G4 and G5) must be reported in figure 5 and 9.

CV. The location of the M2, G3, G4 and G5 core sites has now been added in Fig. 1 and Table 1. Temperatures at 10 m for the core sites were included in Table 1. When plotting the individual M2, G3, G4 and G5 accumulation and d18O profiles in fig. 5 and 9, respectively, the individual lines were hard to visualize. We decided to plot the individual profiles in an additional figure that has been included as supplementary data (Fig. S1).

10. SMB of KC, S100 and Composite appear similar, whereas the KM and BI are similar, but significant different, clearly the two groups represent different climate condition and history, authors must provide some explanation in the discussion.

[Figure]

CV. We have now included more details regarding the point noted by the referee, please see response to point 16 in this letter for a complete overview of the added text.

….Clearly ice rises present different SMB and isotope, the KC SMB is "medium/high" for Antarctic standard condition, and with this value normally the seasonal signal is well preserved (see WAIS), the absence of annual signal is related at others factor that must be analysed.

CV. This is now discussed in section 4.2 and the first paragraph of section 5 has been modified, accordingly (for more details, please refer to answers to points 11 and 13 of this letter).

11. KC isotope profiles dO18 and deuterium appear on figure 7 very differents, could be due to the melting? KC profile appears smooth in the lower part of profile, could the Authors comments?

CV. Stratigraphy in the KC core does show ice lenses, however, their thickness and number does not suggest that there is substantial surface melting or percolation at this site, as to account for the lack of a well preserved subannual isotope signal. Most of ice lenses at KC are thinner than 1 cm, with the thickest having 1.5 cm. In terms of ice content per meter of firn, the KC core has in average no more than 3 % of ice per meter during the period 1958–2012, therefore, we hypothesize than most likely a combination of wind scouring and snow redistribution is affecting the subannual leading to the lack of well-preserved seasonal cycles. We have now modified the first paragraph of section 5, accordingly.

12. At pag 7 line 15 it reports stake measurements: how many per each site? Single stake respect to single core? Explain

CV. As part of another study (V. Goel's Ph D thesis), data from an extensive network of stakes is available at the three ice rises. The data provided in this paper is from the stake closest to each coring site. In the case of KC and KM the stakes are relatively far

away from the core drilling site (about 1 km) while for BI, the stake is nearby (about 40 m away from the coring site). We are aware that this is far from optimal and rather than providing the specific data points we choose to provide this more general information in the paper.

The paragraph in page 7 now says: "SMB derived from the stake closest to each core site (40 m to 1 km) at the three ice rises in 2013 are similar to average SMB values from cores at KC (0.22 and 0.24 m w.e. yr-1 from the stake and core data, respectively) and BI (0.73 and 0.70 m w.e. yr-1), but differ at KM (0.38 versus 0.68 m w.e. yr-1). Differences in point estimates for single years are to be expected given the spatial variability of snow accumulation. The spatial variability of SMB on the ice rises from stake and GPR data will be presented elsewhere."

13. Pag 7 line line 18, on the base of figure 6 the melting occurs several time per year, not "some summers". The importance of melting layer in the stratigraphy must be analysed in more depth with comparison of position respect to nssSO4, isotope and MSA, in particular for KC cores. The site present a "relatively high" SMB (0.24 m we yr) and it is anomalous that the seasonal stratigraphy is not preserved. Could be the combination of wind scouring and melting obliterated the seasonal signal?

CV. We agree with Ref. #2 in that the stratigraphy profile needed a deeper discussion. Unfortunately, we do not have notes on the ice lenses thickness and number in the KM and BI cores, therefore, we only report the stratigraphy for BI. Consequently, we have now added MSA and nssSO42- in Fig.6 (c-d, respectively), and the paragraph in section 4.2 has been modified as follows:

Page 7: "In all three cores, there are ice layers of varying thickness, indicating that melt occurs several times per year; we have no evidence, however, for mass transport between annual layers. Figure 6 shows the number of ice lenses and thickness related to density, d18O, MSA and nssSO42- concentrations in the KC core. There is no direct correspondence between SMB, d18O and the ice layers in the core from KC (Figure 6),

in agreement with what has been shown previously from the core S100 (Kaczmarska et al., 2006). We compare melt features to the MSA and nssSO42- profiles in the KC core, but do not find a systematic association between ice lenses and anomalies in the MSA or nssSO42- concentrations, as we could expect from redistribution of ions by meltwater percolation and refreezing. Some correspondence exists between the thickest ice layers and peaks in the nssSO42- record (e.g. at 21 m, 20 m, 18 m, and 13 m, Figure 6) but there are no such peaks in the MSA record, as would be expected for an ion that it is just as readily eluted as nssSO42-. Therefore, while redistribution of ions by meltwater cannot be ruled out, it is not likely a dominant post-depositional effect that would significantly influence the seasonal isotopic or chemical signals at the core sites. It is more likely that the development of ice lenses is a local process depending on several factors, including air and snow pack temperatures, and that the combination of post-depositional processes, such as wind scouring, contribute to the perturbation of the sub-annual signal in the KC core site."

In addition, paragraph 1 in section 5 has been modified, accordingly.

14. Pag 9 if the data are not significant at 95% and it is not reported in figure 7, I do not understand their importance.

CV. In view that the d data presented here is the first data set for the region, we consider that it is important to mention this information, even though not significant, in order to set the most complete picture possible for the interpretation of ice/firn cores that could be eventually retrieved at FIS in the future. However, we are open to remove this information from the final version of the manuscript if still considered unnecessary.

15. Pag 10 line 32, the topographical and post-depositional effects are not sufficient presented in section 4.2

CV. We have now included a more detailed presentation of the post-depositional effects in section 4.2 (please refer to point 13 of this letter). We have also added 50-m contours at each ice rise which allows a better understanding of the topographic effects affecting

the SMB described in section 4.2. Regarding topographical effects on SMB, please refer to the answer to point 16 in this letter.

16. The different trend in SMB between ice shelf and ice rises must be also analysed respect the observation of the behaviour of SMB Antarctica proposed by Frezzotti et al., 2013.

CV. The referee is right regarding this point and consequently, we have added a more detailed discussion of the topographical effects on SMB and the temporal SMB trends in section 4.2. The added paragraphs say:

On topographical effects:

"Our results suggest that the differences in accumulation at KM and BI compared to KC and the other core sites at FIS, is most likely related to topographical effects. This can be further explored by referring to the study by Altnau et al. (2015) which presents a vast coverage of SMB and d18O for coastal and inland DML. By inspecting Figure 2 in Altnau et al. (2015), it can be observed that high annual SMB values, similar to those measured at the KM and BI sites, occur in few locations only associated with pronounced topographic features, e.g. mountain ranges and troughs, i.e. anything where orographic lift may induce precipitation in comparison to the flat areas in the proximities."

On temporal SMB trends:

"Frezzotti et al. (2013) investigated Antarctic SMB over the last 800 years, and found that there was statistically non-significant changes in SMB over most of Antarctica, with no overall clear temporal trend over the longest timescale. However, they also report a clear increase in SMB (>10 %) since the 1960s in regions where the SMB is high, i.e. coastal regions, and over the highest part of the East Antarctic ice divide. The authors attribute these dissimilar trends between high-SMB locations and the rest of Antarctica to a higher frequency of blocking anticyclones. These anticyclones increase precipita-

tion at coastal sites and lead to the advection of moist air at the highest areas. Strong winds producing snow redistribution and erosion would account for the reduction on SMB at windy sites. As discussed above, our results show that the SMB trends at KC is similar to SMB trends reported elsewhere at FIS and western DML (Isaksson and Melvold, 2002; Kaczmarska et al., 2004; Divine et al., 2009; Schlosser et al., 2014). No significant temporal trends in SMB are found at KM and BI."

Since the KM and BI records are rather short and reveal a high interannual SMB variability and no significant trend on SMB during the period that they cover 1995(96)-2014; therefore, we preferred not to venture further regarding this finding in order to not over interpret the results. We state this in section 5 as: "Due to the restricted length of the KM and BI cores, further analysis of the spatial and temporal differences of SMB and water stable isotopes at these ice rises in a climatic context would be speculative."

17. In the figure 1 should reported the contour line at 10 m.

CV. We have now included 50-m contours at the three ice rises. We consider that 10-m contours would be graphically untenable at this scale. The MOA image gives an excellent impression of the overall form of the ice rises and together with the 50-m contours, we consider the figure has enough detail now to support the discussion in this manuscript.

References

Altnau, S., Schlosser, E., Isaksson, E., and Divine, D.: Climatic signals from 76 shallow firn cores in Dronning Maud Land, East Antarctica. The Cryosphere, 9, 925-944, doi: 10.5194/tc-9-925-2015, 2015.

Cole-Dai J, Mosley-Thompson E. and Thompson L. G.: Quantifying the Pinatubo volcanic signal in south polar snow, Geophys. Res. Lett., 24(21), 2679–2682, doi: 10.1029/ 97GL02734), 1997.

Dibb J. E., and Whitlow, S. I.: Recent climate anomalies and their impact on snow

chemistry at South Pole, 1987–1994, Geophys. Res. Lett., 23(10), 1115–1118, 1996.

Divine, D. V., Isaksson, E., Kaczmarska, M., Godtliebsen, F., Oerter, H., Schlosser, E., Johnsen, S. J., van den Broeke, M., van de Wal, R. S. W.: Tropical Pacific-high latitude south Atlantic teleconnections as seen in d18O variability in Antarctic coastal ice cores, J. Geophys. Res., 114, D11, D11112, doi: http://dx.doi.org/10.1029/2008JD010475, 2009.

Frezzotti, M., Scarchilli, C., Becagli, S., Proposito, M, and Urbini, S.: A synthesis of the Antarctic surface mass balance during the last 800 yr, The Cryosphere, 7, 303–319, doi: 10.5194/tc-7-303-2013, 2013.

Isaksson, E. and Melvold, K.: Trends and patterns in the recent accumulation and oxygen isotopes in coastal Dronning Maud Land, Antarctica: interpretations from shallow ice cores, Ann. Glaciol., 35, 175–180, 2002.

Kaczmarska, M., Isaksson, E., Karlöf, L., Brandt, O., Winther, J.-G., van de Wal, R. S. W., van de Broeke, M., and Johnsen, S. J.: Ice core melt features in relation to Antarctic coastal climate, Antarctic Science, 18(2), 271–278, doi: 10.1017/S0954102006000319, 2006.

Kaczmarska, M., Isaksson, E., Karlöf, L., Winther, J-G., Kohler, J., Godtliebsen, F., Ringstad Olsen, L., Hofstede, C. M., Van Den Broeke, M. R., Van De Wal, R. S.W., Gundestrup, N.: Accumulation variability derived from an ice core from coastal DML, Antarctica, Ann. Glaciol. 39, 339–345, 2004.

Schlosser, E., Anschütz, H., Divine, D., Martma, T., Sinisalo, A., Altnau, S., and Isaksson, E., Recent climate tendencies on an East Antarctic ice shelf inferred from a shallow firn core network, J. Geophys. Res. Atmos., 119, 6549–6562, 2014.

Schlosser, E., Duda, M. G., Powers, J. G, Manning, K. H.: The precipitation regime of Dronning Maud Land, Antarctica, derived from AMPS (Antarctic Mesoscale Prediction System) Archive Data. J. Geophys. Res., 113. D24108, doi: 10.1029/2008JD009968,

2008.

Vega, C. P.: Análisis e interpretación de las concentraciones de anions en testigos de hielo de la isla James Ross, Península Antártica, Thesis at the University of Chile, Facultad de Ciencias Químicas y Farmacéuticas, Santiago, www.cybertesis.uchile.cl/tesis/uchile/2008/qf-vega_c/pdfAmont/qf-vega_c.pdf, 2008.

Please also note the supplement to this comment:
http://www.the-cryosphere-discuss.net/tc-2016-164/tc-2016-164-AC1-supplement.pdf

**Supplement:**

Supplementary data

**Surface mass balance and water stable isotopes derived from firn cores on three ice rises, Fimbul Ice Shelf, Antarctica**

Carmen P. Vega,[1,2] Elisabeth Schlosser,[3,4] Dmitry V. Divine,[1] Jack Kohler,[1] Tõnu Martma,[5] Anja

5   Eichler,[6] Margit Schwikowski[6], and Elisabeth Isaksson[1]

[1]Norwegian Polar Institute, N-9296 Tromsø, Norway
[2]Department of Earth Sciences, Uppsala University, Villavägen 16, SE 752 36, Uppsala, Sweden
[3]Institute of Atmospheric and Cryospheric Sciences, University of Innsbruck, Innsbruck, Austria
[4]Austrian Polar Research Institute, Vienna, Austria
10   [5]Institute of Geology, Tallinn University of Technology, Tallinn, Estonia
[6]Paul Scherrer Institute, 5232 Villigen PSI, Switzerland

*Correspondence to*: Carmen P. Vega (carmen.vega@geo.uu.se)

[Figure]

15

**Figure S1:** a) Annual SMB, and b) mean annual $\delta^{18}O$ for M2, G3, G4, G5 and the composite FIS core record (Schlosser et al., 2014). The dashed red line indicates the linear regression in the composite core for the period (1983–2009) shown in Figure 5 and 9 for SMB and $\delta^{18}O$, respectively.